# Absence of Heme Oxygenase-1 Affects Trophoblastic Spheroid Implantation and Provokes Dysregulation of Stress and Angiogenesis Gene Expression in the Uterus

**DOI:** 10.3390/cells13050376

**Published:** 2024-02-22

**Authors:** Maria Laura Zenclussen, Sina Ulrich, Mario Bauer, Beate Fink, Ana Claudia Zenclussen, Anne Schumacher, Nicole Meyer

**Affiliations:** 1Instituto de Salud y Ambiente del Litoral (ISAL, UNL-CONICET), Facultad de Bioquímica y Ciencias Biológicas, Universidad Nacional del Litoral (UNL), Santa Fe 3000, Argentina; 2Experimental Obstetrics and Gynecology, Medical Faculty, Otto-von-Guericke University Magdeburg, 39108 Magdeburg, Germanyana.zenclussen@ufz.de (A.C.Z.); anne.schumacher@ufz.de (A.S.); 3Department of Environmental Immunology, Helmholtz Centre for Environmental Research-UFZ, 04318 Leipzig, Germany; mario.bauer@ufz.de (M.B.); beate.fink@ufz.de (B.F.); 4Perinatal Immunology, Saxonian Incubator for Clinical Translation (SIKT), Medical Faculty, Leipzig University, 04103 Leipzig, Germany

**Keywords:** heme oxygenase-1, implantation, pregnancy, angiogenesis

## Abstract

The enzyme heme oxygenase-1 (HO-1) is pivotal in reproductive processes, particularly in placental and vascular development. This study investigated the role of HO-1 and its byproduct, carbon monoxide (CO), in trophoblastic spheroid implantation. In order to deepen our understanding of the role of HO-1 during implantation, we conducted in vivo experiments on virgin and pregnant mice, aiming to unravel the cellular and molecular mechanisms. Using siRNA, HO-1 was knocked down in JEG-3 and BeWo cells and trophoblastic spheroids were generated with or without CO treatment. Adhesion assays were performed after transferring the spheroids to RL-95 endometrial epithelial cell layers. Additionally, angiogenesis, stress, and toxicity RT2-Profiler™ PCR SuperArray and PCR analyses were performed in uterine murine samples. HO-1 knockdown by siRNA impeded implantation in the 3D culture model, but this effect could be reversed by CO. Uteruses from virgin *Hmox1^−/−^* females exhibited altered expression of angiogenesis and stress markers. Furthermore, there was a distinct expression pattern of cytokines and chemokines in uteruses from gestation day 14 in *Hmox1^−/−^* females compared to *Hmox1^+/+^* females. This study strongly supports the essential role of HO-1 during implantation. Moreover, CO appears to have the potential to compensate for the lack of HO-1 during the spheroid attachment process. The absence of HO-1 results in dysregulation of angiogenesis and stress-related genes in the uterus, possibly contributing to implantation failure.

## 1. Introduction

Heme oxygenase-1 (HO-1), encoded by the *HMOX1* gene [1], is an enzyme involved in plenty of physiological processes [2]. It catalyzes the degradation of free heme into carbon monoxide (CO), free iron (Fe^2+^) and biliverdin [3]. HO-1 is ubiquitously expressed in most tissues across multiple species [4] and plays a crucial physiological protective role [5]. Various stimuli, such as heme, heat shock, heavy metals and cytokines, induce this enzyme in response to oxidative stress [3]. In humans, only a few cases of *HMOX1* deficiency have been reported [6,7,8,9,10], leading to complications like generalized inflammation, disturbances in the coagulation/fibrinolysis system and vascular endothelial injury [7,11].

The significance of HO-1 in reproductive processes has been documented in both humans and rodents. Polymorphisms in the HO-1 promoter or a partial deficiency in maternal HO-1 are associated with pregnancy complications, including recurrent miscarriages, pre-eclampsia and fetal growth restriction [12,13,14,15,16]. Existing studies on HO-1 during pregnancy often focused on its contribution to placental development [17], as defective placentation is linked to the mentioned complications. A study by Zhao et al. [18] on ton heterozygous (*Hmox1^+/−^*) placentas revealed less efficient vascular adaptation, with maternal HO-1 levels determining defects in the placental vasculature, not the fetal genotype. Other research indicated HO-1’s crucial role in modulating the angiogenic/anti-angiogenic balance in placental vascular homeostasis [19]. Notably, in murine models, application of CO had a pro-angiogenic effect in the placenta [20,21] and was able to mitigate the effects caused by *Hmox-1* deficiency [22].

Despite the undeniable importance of HO-1 in the placental structural and vascular development, our group’s previous work suggested its crucial role during blastocyst implantation. Using an in vitro implantation model involving the co-culture of murine blastocysts with autologous uterine epithelial cells [22], we observed that the blastocysts from the *Hmox1^+/−^* X *Hmox1^+/−^* or *Hmox1^+/^*^−^ X *Hmox1^−/−^* combinations had reduced attachment to uterine epithelial cells compared to the control *Hmox1^+/+^* X *Hmox1^+/+^* combination. Notably, the non-attached blastocysts were proven to be *Hmox1^−/−^* [22]. These findings strongly suggest that HO-1 may play a crucial role in the implantation process.

The potential of CO, a byproduct of HO-1, to improve implantation success in the absence of HO-1 remains unknown. Consequently, we aimed to explore this aspect using an in vitro approach. To further investigate the importance of HO-1 during implantation, our focus shifted to elucidating the cellular and molecular mechanisms underlying our previous observations. We hypothesized that the lack of HO-1 provokes dysregulation of other transcripts in the uterine environment, contributing to implantation failure. In conclusion, this study aimed to enhance our understanding of how HO-1 may affect implantation. 

## 2. Materials and Methods

### 2.1. Cell Culture

The two trophoblastic human cell lines JEG-3 and BeWo, as well as the epithelial endometrial cell line RL-95, were purchased from the American Type Culture Collection (ATCC, Manassas, VA, USA) and maintained under standard culture conditions in 5% CO_2_ at 37 °C. The culture media used were Dulbecco’s modified Eagle’s medium (DMEM) F12 (Thermo Fisher Scientific, Waltham, MA, USA) for the BeWo and RL-95 cells, and DMEM (Thermo Fisher Scientific) for the JEG-3 cells. Each medium was supplemented with 1% penicillin–streptomycin (P/S; Biowest, Nuaillé, France) and 10% fetal bovine serum (FBS; PAN-Biotech, Aidenbach, Germany).

### 2.2. HO-1 siRNA Transfection of Trophoblast Cells Using Electroporation

Either 0.2 nM Silencer^®^ Select HO-1 siRNA (ID: s6674 from Ambion/Thermo Fisher Scientific) or 0.2 nM Silencer^®^ Select Negative Control siRNA (internal control) was mixed with 4 × 10^5^ JEG-3 or BeWo cells in 400 µL OptiMEM medium (Thermo Fisher Scientific) in electroporation cuvettes. The cells were pulsed using a square wave protocol of the Gene Pulser electroporator (BioRad, Feldkirchen, Germany) at a voltage of 140 V for 50 ms. Afterwards, the pulsed cells were immediately transferred into 6-well plates or into 35 mm petri dishes, each containing 2 mL of medium for Western blot analyses or trophoblastic spheroid generation, respectively.

### 2.3. Western Blot Analysis of siRNA Transfection Efficacy

HO-1 knockdown after siRNA transfection was confirmed by Western blot analysis after 48 h of culture. The JEG-3 (passage 35–38) or BeWo (passage 20–23) cells were lysed using lysis buffer (containing 10% NP-40 detergent, 0.1 mg/mL n-dodecil beta maltoside, 10 mM sodium monovanadate, 100 mM phenylmethylsulfonyl fluoride, 500 mM sodium fluoride, 500 mM ethylenediaminetetraacetic acid, 5 M sodium chloride and 1 M Tris) and the proteins were extracted. The total protein was quantified using the BCA kit (Thermo Fisher Scientific). Then, 30 µg of protein was loaded on a 10% sodium-dodecyl-sulfate polyacrylamide gel and subsequently transferred to a polyvinylidene difluoride membrane (Amersham Biosciences, Buckinghamshires, England). After blocking with 5% milk powder in tris-buffered saline, the membrane was incubated with primary antibodies against HO-1 (rabbit; 1:500, Santa Cruz Biotechnology, Santa Cruz, CA, USA) and beta-actin (rabbit; 1:1000, Santa Cruz Biotechnology, Santa Cruz, CA, USA) at 4 °C overnight. On the next day, the membrane was washed in tris-buffered saline and incubated with a goat anti-rabbit secondary antibody (horseradish-conjugated; 1:1000, Abcam, Cambridge, MA, USA) for 1 h at room temperature. The HO-1 and beta-actin protein expression was finally visualized using chemiluminescence on a BioRad imaging system. 

### 2.4. Trophoblastic Spheroid Generation and Adhesion Assay

To generate trophoblastic spheroids, 4 × 10^5^ siRNA-treated and non-treated JEG-3 (*n* = 8 independent assays) or BeWo (*n* = 5 independent assays) cells (controls) were cultured in 35 mm petri dishes on a gyratory shaker (70 rev/min, IKA, Staufen, Germany) to prevent cell adhesion and enable spheroid formation. For the CO treatment (500 ppm) during spheroid formation, JEG-3 cells (*n* = 5 independent assays) were cultured in 2 mL alternative medium (without FBS) to create non-adherent conditions as a gyratory shaker could not be inserted into the CO chambers (BioSpherix, Parish, NY, USA). In a previous study, we proved 500 ppm CO to be an effective dose to overcome the harmful effects in trophoblast cells that were initiated by HO-1 deficiency [22]. Unfortunately, the BeWo cells did not form spheroids in the alternative medium; thus, they could not be used for the CO treatment experiments. For both experimental setups, after 5 days of culture, the spheroids were harvested and transferred to a confluent layer of RL-95 endometrial epithelial cells (10 spheroids per well/assay). The day before, 4 × 10^4^ RL-95 cells were plated in 96-well plates to allow cell adhesion to the culture plate. The trophoblastic spheroids were then allowed to attach to the endometrial epithelial cells for 24 h. Attachment of the spheroids was proven by overhead centrifugation at 25× *g* at room temperature for 5 min followed by enumeration of the attached spheroids under light microscopy (Mikroskop Axiovert 40C, Zeiss Microimaging, Jena, Germany). 

### 2.5. Mice and Sample Collection

*Hmox1^+/+^* and *Hmox1^−/−^* BALB/c mice were kindly provided by Dr. Saw Feng-Yet [23]. The genotype of the mice was analyzed by PCR using the following primers: E3/I3R (5′-GGTGACAGAAGAGGCTAAG-3′ and 5′-CTGTAACTCCACCTCC AAC-3′) and Neo1/E4 (5′-TCTTGACGAGTTCTTCTGAG-3′ and 5′-ACGAAGTGACGCCATCTGT-3′). The E3/I3R set of primers amplifies a 456 bp fragment of the wild-type allele, whereas the Neo1/E4 amplifies a 400 bp fragment of the mutated allele. The uteruses from *Hmox1^+/+^* (*n* = 7) or *Hmox1^−/−^* females (*n* = 8) used for the SuperArray assays were obtained from an experimental approach previously published [24]. The uteruses were snap frozen in liquid nitrogen and kept at −80 °C. 

Eight- to eleven-week-old female *Hmox1*^+/+^ (*n* = 10) and *Hmox1*^−/−^ (*n* = 6) were mated with *Hmox1*^+/+^ and *Hmox1*^−/−^ males and maintained at the animal barrier facility of Magdeburg University. The mice were kept with a 12 h light/dark cycle at 22 ± 2 °C and air humidity of 40–60%. The mice received water/food ad libitum. The animal experiments were performed according to the institutional guidelines upon ministerial approval (Landesverwaltungsamt Sachsen Anhalt: 42502-2-1327 Uni MD). All the experiments were conducted by authorized persons according to the Guide for Care and Use of Animals in Agriculture Research and Teaching.

The appearance of the vaginal plug indicated day 0 of pregnancy (gd0). The animals were sacrificed at gd14, and the uterus tissue was snap frozen in liquid nitrogen and stored at −80 °C. The samples were employed to perform quantitative PCR analysis.

### 2.6. Mouse Angiogenesis RT2-Profiler™ PCR Array

The Mouse Angiogenesis RT2-Profiler™ PCR Array (APMM-024; SuperArray Bioscience, Frederick, MD, USA) profiles the expression of 84 genes involved in angiogenesis as well as 5 reference genes (actin, beta, *Actb*; glyceraldehyde-3-phosphate dehydrogenase, *Gapdh*; heat shock protein 90 kDa alpha (cytosolic), class B member 1, *Hsp90ab1*; hypoxanthine guanine phosphoribosyl transferase 1, *Hprt1*; and glucuronidase, beta, *Gusb*) by real-time PCR using the SYBR Green detection method. First, a pool of RNA from the uteruses of wild-type females (*Hmox1^+/+^*, *n* = 8) was generated, as well as a pool of RNA from the uteruses of *Hmox1^−/−^* females (*n* = 7). 

Thereafter, the RNA concentration of the pools was measured, and 1 μg of total RNA from each pool was reverse transcribed using the RT^2^ PCR Array First Strand Kit (C-02, SuperArray Bioscience, Frederick, MD, USA). The generated cDNA was diluted with an appropriate volume of instrument-specific 2x SuperArray RT2 Real-Time™ SYBR Green PCR Master Mix (PA-012, SuperArray Bioscience, Frederick, MD, USA) and ultra-pure water, and 25 µL of this reaction mix was added to each well of the PCR array. The real-time PCR reaction was performed in an iCycler iQ™ Real-Time PCR Detection System applying the following program: 10 min at 95 °C and 40 cycles of 15 s at 95 °C and 1 min at 60 °C. The iCycler iQ™ Real-Time PCR Detection System was used to calculate the Ct value for each well. This array was performed for the pool of *Hmox1^+/+^* uteruses as well as for the pool of *Hmox1^−/−^* uteruses. Data were normalized to three reference genes (*Hprt1*, *Gapdh* and *Actb*) and analyzed by the comparative Ct method (2^−ΔΔCT^), using the *Hmox1^+/+^* pool as the control sample and the *Hmox1^−/−^* pool as the test sample. The expression levels were normalized to the mean of the reference genes for each pool of samples. 

The names of the analyzed genes are listed in the Appendix A.

### 2.7. Mouse Stress and Toxicity PathwayFinder™ RT^2^ Profiler™ PCR Array

The Mouse Stress and Toxicity PathwayFinder™ RT^2^ Profiler™ PCR Array (APMM-003A; SuperArray Bioscience, Frederick, MD, USA) profiles the expression of 84 genes whose expression level is indicative of stress and toxicity. Five reference genes are also included (actin, beta, *Actb*; glyceraldehyde-3-phosphate dehydrogenase, *Gapdh*; heat shock protein 90kDa alpha (cytosolic), class B member 1, *Hsp90ab1*; hypoxanthine guanine phosphoribosyl transferase 1, *Hprt1*; and glucuronidase, beta, *Gusb*). As explained for the Angiogenesis RT2-Profiler™ PCR Array, the analysis was performed by real-time PCR using the SYBR Green detection method, using the *Hmox1^+/+^* pool as the control sample and the *Hmox1^−/−^* pool as the test sample. The real-time PCR reaction was performed in an iCycler iQ™ Real-Time PCR Detection System applying the following program: 10 min at 95 °C and 40 cycles of 15 s at 95 °C and 1 min at 60 °C. The iCycler iQ™ Real-Time PCR Detection System was used to calculate the Ct value for each well. This array was performed for the pool of *Hmox1^+/+^* uteruses as well as for the pool of *Hmox1^−/−^* uteruses. The expression levels were normalized to the mean of the reference genes (*Hprt1*, *Gapdh* and *Actb*) for each pool of samples. 

The names of the analyzed genes are listed in Appendix A.

### 2.8. RNA Isolation, cDNA Synthesis and Quantitative Real-Time PCR

RNA isolation: The total RNA was isolated using TRIzol^®^ reagent according to the manufacturer’s instructions. Briefly, frozen uterine tissue was homogenized in TRIzol^®^ with an Ultra Turrax T25 homogenizer (NeoLab, Heidelberg, Germany). The total RNA was extracted with chloroform, precipitated with isopropanol, washed with ethanol, and diluted in RNA-free water. Finally, RNA quantification was performed by measuring the UV absorbance at 260 nm, and a quality check was performed by 280 nm measurement. Finally, the RNA was stored at −80 °C.

cDNA synthesis: 2 µg RNA was incubated with RNAse-free water and oligo dTs at 75 °C (10 min) and afterwards on ice (2 min). The marked mRNA was incubated with dNTPs (2.5 mmol/mL) and DNAse (2 U/µL) in a reaction buffer at 37 °C (30 min). DNAse was inactivated at 75 °C (5 min) and the samples were incubated on ice (2 min). After the addition of reverse transcriptase (200 U/µL) and RNAse inhibitor (40 U/µL), cDNA was synthesized at 42 °C (60 min). The reverse transcriptase was inactivated at 94 °C (5 min) and the cDNA was stored at −20 °C. 

Real-time polymerase chain reaction (RT-PCR): RT-PCR was performed with FastStart Universal Probe Master Mix (Roche, Vienna, Austria) on the BioMark HD system (Fluidigm, San Francisco, CA, USA) using EvaGreen DNA-binding Dye with BioMark™ 48.48 Dynamic Array Integrated Fluidic Circuits according to the manufacturer’s recommendations. All the reactions were performed in duplicates. Actin beta (*Actb*), glyceraldehyde-3-phosphate dehydrogenase (*Gapdh*), ribosomal protein, large, P0 (*Rplp0*), and ubiquitin-C (*Ubc*) were used as reference genes. To quantify the relative expression, the expression of the gene of interest was normalized to the geometric mean of four reference genes. This was performed individually for each sample. All the qPCR primers used are listed in Appendix A. 

### 2.9. Statistical Analysis

The normality of the datasets was assessed with the Shapiro–Wilk test. Accordingly, the data were analyzed with either parametric or non-parametric tests. The data are presented as medians or means depending on their distribution. The number of samples or mice is indicated in the materials/methods part. The statistical tests and the *p* values are indicated in the figure legends. The statistical analyses were performed with GraphPad Prism 9.0.

## 3. Results

### 3.1. HO-1 Knockdown by siRNA Impairs Implantation in an In Vitro 3D Culture Model

We analyzed implantation using an in vitro approach after taking into consideration that *Hmox1^−/−^* animals are not able to successfully implant after natural mating. To overcome this limitation, we employed an in vitro implantation model, using a 3D culture of human cell lines. The implantation analysis involved culturing trophoblastic spheroids over a RL-95 endometrial epithelial cell monolayer. Two trophoblast cell lines, BeWo and JEG-3, were used for the analysis. In both cell lines, HO-1 was knocked down by siRNA transfection. The transfection efficiency was checked by the analysis of HO-1 protein expression by Western blot (Appendix A). 

Once the knockdown of HO-1 expression was achieved, the spheroids were transferred over a monolayer of endometrial epithelial cells, and the attachment was analyzed after 24 h. Interestingly, the attachment was significantly impaired when HO-1 was knocked down in the BeWo spheroids (Figure 1a) and JEG-3 spheroids (Figure 1b). More precisely, attachment of the BeWo spheroids was reduced from 97.60% of control cells and 73.40% of mock siRNA-treated cells to 10.20% when HO-1 siRNA was applied (Figure 1a). For the JEG-3 cells, 91.63% of control cells and 83.50% of mock siRNA-treated cells attached to the endometrial epithelial cells, while this was true only for 11.25% of HO-1 siRNA-treated cells (Figure 1b).

To rule out the possibility that the observed impairment in the spheroid attachment was due to a direct effect of the treatment on the cell viability, number of spheroids and/or cell migration, these parameters were analyzed in all the conditions assayed. Interestingly, HO-1 knockdown did not affect any of these parameters (Appendix A). 

Our in vitro results confirm the hypothesis that HO-1 is essential for implantation, because its knockdown leads to the compromised attachment of trophoblasts to endometrial epithelial cells.

Subsequently, we aimed to investigate whether the HO-1 absence effect on spheroid attachment could be reverted by one of the metabolites of the HO-1 reaction, namely carbon monoxide (CO). For that purpose, we performed the experiment with JEG-3 spheroids in the presence of CO. 

We confirmed our initial observations proving that HO-1 knockdown in JEG-3 spheroids by means of HO-1 siRNA reduced spheroid attachment to 17.00% compared to the 55.00% attachment rate after mock siRNA treatment or to the 94.33% attachment rate without siRNA treatment (Figure 2). Interestingly, when cells were in the presence of 500 ppm CO, the effect observed due to HO-1 knockdown was reverted (Figure 2), as the attachment rates for the spheroids without siRNA, with mock siRNA and even with HO-1 siRNA were 95.33%, 92.50% and 94.00%, respectively, in the presence of CO. This indicates that CO may be sufficient to overcome the lack of HO-1 in the process of spheroid attachment.

### 3.2. Uteruses from Hmox1^−/−^ Females Show Altered Expression of Angiogenesis and Stress Markers

We next aimed to analyze which molecules are differently expressed in vivo as a consequence of HO-1’s absence. For this, we used uterine tissue from *Hmox1^−/−^* vs. *Hmox1*^+/+^ mice. The Mouse Stress and Toxicity PathwayFinder™ RT^2^ Profiler™ PCR Array showed a strong downregulation of the RAD50 double strand break repair protein (*Rad50*, 1867-fold down-regulation) and glutathione-S-transferase M3 (*Gstm3*; 9-fold down-regulation) in *Hmox1^−/−^* uteruses when compared to *Hmox1^+/+^* uteruses. As expected, no expression of *Hmox1* was found in *Hmox1^−/−^* uteruses. Other genes that showed a >2-fold downregulation were: *Xrcc4*, *Il1*, *Cyp7a1*, *Cyp2b9*, *Cyp2c29*, *Cyp3a11*, *Cyp2a5*, *Cyp1a1* and *Csf2* (Figure 3).

When analyzing the Mouse Angiogenesis RT2-Profiler™ PCR Array, a strong upregulation of *Epas1* (776-fold up-regulation), as well as a >2-fold downregulation of *Fgf6*, *Lect1*, *Leptin*, *Mdk*, *Plxdc1* and *Vegfc*, in *Hmox1^−/−^* uteruses when compared to *Hmox1^+/+^* uteruses were found (Figure 4).

### 3.3. Uteruses from gd14 Hmox1^−/−^ Females Show Altered Expression of Cytokines and Chemokines

At gestation day 14, uteri from both *Hmox1^+/+^* and *Hmox1^−/−^* female mice were collected, and the expression of various genes was compared between the two groups. The analysis of the gene expression showed elevations in interleukins (*Il1b*, *Il4*, *Il6*, *Il11*, *Il27* and *Ifng*; Figure 5a), chemokines (*Ccl2*, *Ccl3*, *Ccl4*, *Cxcl5*, *Cxcl12* and *Cxcl13*; Figure 5b), and matrix metalloprotease 9 (*Mmp9*, Figure 5c). The lack of *Hmox1* induced also a significant upregulation in the gene expression of plasminogen activator inhibitor-1 (*Serpine1*) as well as the expression of transformation-related protein 53 (*Trp53)* (Figure 5c).

## 4. Discussion

The results shown in this work strongly support the hypothesis that HO-1 is essential for successful implantation of spheroids. The knockdown of HO-1 using siRNA in trophoblastic spheroids, either in BeWo or in JEG-3 cells, impaired their ability to implant in a monolayer of RL-95 endometrial epithelial cells. This is in accordance with our previous observation of impaired blastocyst implantation of *Hmox1^−/−^* blastocysts using an in vitro implantation model with blastocysts from the *Hmox1^+/−^* x *Hmox1^+/−^* mouse combination [22]. Notably, in the present study, the knockdown was specifically performed in the cells forming the spheroids, not in the endometrial epithelial cells. This suggests that, even if the endometrial epithelial cells express HO-1, the knockdown of HO-1 in the spheroids alone is sufficient to alter the environment, hindering the implantation process. Remarkably, when the spheroids were cultured in an incubator containing 500 ppm CO, the implantation capacity of the spheroids treated with HO-1 siRNA was reestablished. This suggests that CO, one of the byproducts of the HO-1 reaction, may mediate the implantation process. Many studies propose that the effects attributed to HO-1 are, in fact, due to the products of its enzymatic reaction [25]. The results obtained in this in vitro model of spheroid implantation reinforce this argument.

As HO-1 has a protective role in inflammatory processes, animals deficient in HO-1 (*Hmox1^−/−^* animals) have multiple disturbances, including oxidative damage, tissue injury, chronic inflammation [26], and a Th1-weighted shift in the cytokine responses [27]. Reproductive problems are also evident, with existing research predominantly focusing on the significant role of HO-1 in placentation [17]. Taking into account our previous observation of the involvement of HO-1 in blastocyst implantation, we postulate that HO-1 is also essential for successful implantation. For this reason, we aimed to analyze different marker transcripts related to stress, angiogenesis and inflammation specifically in the uteruses of *Hmox1^−/−^* animals. This approach aims to establish a connection between the absence of HO-1 and a dysregulated uterine environment.

The analysis of *Hmox1^+/+^* and *Hmox1^−/−^* uteruses from virgin animals showed a strong downregulation of *Rad50* in *Hmox1^−/−^* animals. *Rad50* is involved in the regulation of DNA demethylation [28] and in the detection and repair of DNA double strand breaks [29]. To the best of our knowledge, there are no existing reports linking HO-1 with *Rad50*, making this an intriguing avenue for future studies. The 9-fold downregulation of *Gstm3* in *Hmox1^−/−^* uteruses is noteworthy, as a simultaneous downregulation of *Hmox1* and *Gstm3* has been previously reported in mouse liver samples exposed to an omega-6 rich emulsion [30]. Although there are no known works relating *Hmox1* and *Gstm3* expression in the uterus, *Gstm3* expression in the uterine environment was previously described in sheep, where it is expressed predominantly in the epithelial endometrium [31]. As part of the GST family, *Gstm3* is involved in the uptake and detoxification of endogenous and xenobiotic compounds, thereby protecting cells from oxidative DNA damage [32]. The downregulation of *Gstm3* in *Hmox1^−/−^* uteruses suggests a potential predisposition of the epithelia to oxidative damage. This indicates that the antioxidant system in the uterus is altered in the absence of HO-1, not only because of its absence but also for the simultaneous downregulation of other antioxidant genes such as *Gstm3*.

Furthermore, several genes belonging to the Cytochrome P450 (Cyp) family exhibited downregulation in *Hmox1^−/−^* uteruses compared to *Hmox1^+/+^* uteruses. While many of the Cyp genes were downregulated, its downregulation is not so pronounced as for *Cyp2a5*, which was downregulated in *Hmox1^−/−^* uteruses and showed a pronounced decrease by a factor of eight. The *Cyp2a5* gene encodes for Cytochrome P450 2A5, a bilirubin oxidase responsible for microsomal bilirubin degradation [33]. A simultaneous upregulation of murine hepatic *Cyp2a5* and *Hmox1* in oxidative and electrophilic stress has been reported [34]. Considering that one of the byproducts of the HO-1 enzymatic reaction is bilirubin, a downregulation in the expression of *Cyp2a5* is expected when HO-1 is not present.

Regarding the expression of angiogenesis-related genes, a strong upregulation of endothelian PAS domain protein 1 (*Epas1*, 776-fold) was found in *Hmox1^−/−^* uteruses when compared to *Hmox1^+/+^* uteruses. *Epas1* encodes for the HIF2α (hypoxia-inducible factor 2a) and is considered a placenta regulatory gene. Interestingly, elevated expression of *Epas1* has been reported in human samples with intrauterine growth restriction [35] (IUGR). The augmentation of *Epas1* in uterine tissue in *Hmox1^−/−^* animals, along with previous findings in *Hmox1^−/−^* animals exhibiting IUGR [15,16], are in accordance with the finding in human samples.

As for the other genes that are downregulated in *Hmox1^−/−^* uteruses, such as *Fgf6*, *Lect1*, *Lep*, *Mdk*, *Plxdc1* and *Vegfc*, only *Fgf6* and *Lep* show a >2-fold downregulation in *Hmox1^−/−^* uteruses when compared to *Hmox1^+/+^* uteruses. The *Fgf6* gene encodes for fibroblast growth factor 6 (FGF6), which is a member of the FGF family. FGF proteins are potent mitogens that promote angiogenesis [36]. Having found a downregulation in the expression of *Fgf6* in *Hmox1^−/−^* uteruses strongly suggests that angiogenesis may be altered in *Hmox1^−/−^* animals. Surprisingly, there is limited information on FGF6 expression in uteruses or its relation to HO-1, warranting further in-depth investigation.

The 3-fold downregulation of *Lep*, encoding for Leptin, is noteworthy, as Leptin plays a role during pregnancy, being essential for conception, implantation and gestation in mice [37]. The observed alterations in stress as well as in angiogenesis markers strongly support our hypothesis that the lack of HO-1 dysregulates other molecules, exacerbating the challenges to successful implantation.

After finding the mentioned alterations in the stress and angiogenesis markers in *Hmox1^−/−^* uteruses of virgin females, we then analyzed the expression of cytokines and chemokines that play a major role in the uterine environment during gestation. At gd14, we observed altered gene expression of *Il6*, *Ifng*, *Il27*, *Il11*, *Il1b* and *Il4* in *Hmox1^−/−^* uteruses compared to *Hmox1^+/+^* uteruses. Specifically, a significant upregulation of *Il6*, *Ifng*, *Il27*, *Il11*, *Il1b* and *Il4* gene expression was noted in *Hmox1^−/−^* uteruses.

Elevated levels of IL-6 have been previously related to spontaneous abortion [38], preeclampsia [39] and IUGR [40], indicating an inflammatory state. The increased expression of *Il6* in *Hmox1^−/−^* uteruses at gd14 supports the hypothesis of an inflammatory state in the uteruses of *Hmox1^−/−^* animals, potentially detrimental to successful pregnancies. This hypothesis is also reinforced by the increased levels of *Ifn-γ*, typically linked with a pro-inflammatory state during pregnancy complications [41]. *Il27*, correlated with an immune–inflammatory imbalance in the pre-term [42], showed increased expression in *Hmox1^−/−^* uteruses. Similarly, the upregulation of *Il11*, was previously associated with inflammation and pre-eclampsia features in mice [43] and *Il1b*, linked to idiopathic recurrent implantation failure [44], indicated a pro-inflammatory milieu.

IL-4, normally referred to as anti-inflammatory, was augmented in *Hmox1^−/−^* uteruses when compared to *Hmox1^+/+^* uteruses from gd14. Augmented levels of IL-4 do not generally correlate with pregnancy complications, so this result was unexpected. However, taking into account that five pro-inflammatory cytokines were upregulated (*Il6*, *Ifng*, *Il27*, *Il11*, and *Il1b*) and only one anti-inflammatory cytokine (*Il4*) was upregulated, this suggests a prevailing pro-inflammatory environment.

The expression of chemokines in the uteruses of gd14 *Hmox1^+/+^* and *Hmox1^−/−^* females was also analyzed, as an alteration in some chemokines was found in *Hmox1^+/−^* uteruses by other authors [45]. A significant augmentation of various chemokines (*Ccl2*, *Ccl3*, *Ccl4*, *Cxcl5*, *Cxcl12* and *Cxcl13*) was found in *Hmox1^−/−^* uteruses when compared to *Hmox1^+/+^* uteruses. Interestingly, CCL2 was significantly reduced in *Hmox1^+/−^* uteruses, as reported by Zhao et al. [45], in opposition to our result of an augmentation of *CCL2* in the total absence of HO-1. Considering the inflammatory nature of CCL2, CCL3, CCL4 and CXCL5 [46], the altered chemokine expression aligns with the elevated levels of pro-inflammatory cytokines.

Collectively, the in vitro findings from our study strongly emphasize the crucial role of HO-1 and its byproduct CO in spheroid implantation. Regarding the gene expression analyses of uterine tissue, our results strongly suggest that the absence of HO-1 in vivo provokes dysregulation in the expression of angiogenesis and stress-related genes in the uterus. Specifically, *Hmox1^−/−^* uteruses showed a strong upregulation of *Epas1* together with a downregulation of *Fgf6*, *Lep* and *Vegfc* when compared to *Hmox1^+/+^* uteri. Regarding the stress markers, the absence of HO-1 was associated with lower levels of *Rad50*, *Gstm3* and *Cyp2a5*. In addition, the absence of HO-1 predisposes the uterine environment to an inflammatory profile, incompatible with healthy pregnancies. This inflammatory profile was characterized by significantly upregulated levels of *Il1b*, *Il6*, *Il11*, *Il27 and Ifnγ*, as well as the chemokines *Ccl2*, *Ccl3*, *Ccl4*, *Cxcl5. Cxcl12 and Cxcl13*, in *Hmox1^−/−^* uteruses. These results do not diminish the acknowledged importance of HO-1 in the placenta but rather highlight the additional importance of the HO-1 expression in the uterus for earlier pregnancy stages that may condition the initial steps of a successful pregnancy.

A current limitation of this study is the translation potential of our in vitro observations into the in vivo situation. Although our in vitro implantation model provides evidence of a significant contribution of HO-1 in the implantation process, this model cannot mimic all the facets of the in vivo situation in which HO-1 may interact with a variety of factors. Another limitation is the lack of information on the effect of CO on uterine cells in vivo to link the in vitro with the in vivo observations. However, we have carried out previous work using CO in vivo during implantation and early placental development (50 ppm during days 3 to 8 post-coitum) [20,22]. In these experiments, the application of CO in vivo showed both anti-inflammatory and pro-angiogenic effects, together with a reduction in the levels of circulating free heme. Although the analyses were not focused on the uterine tissue, these previous results lead us to speculate that CO application in vivo may compensate for the lack of HO-1 in the uterus by exerting both anti-inflammatory and pro-angiogenic effects. Taking into account our promising results with CO in the in vitro setting, this opens up the possibility of future work investigating the potential of CO as a therapeutic approach to overcome implantation failure.

## Figures and Tables

**Figure 1 cells-13-00376-f001:**
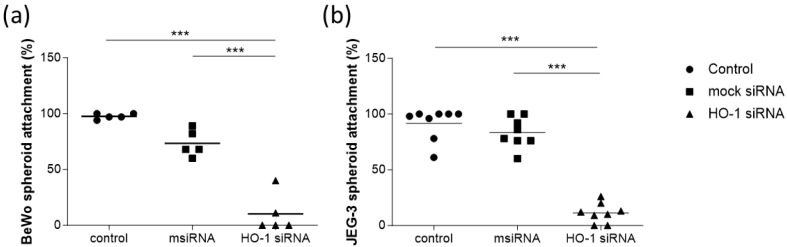
Percentage (%) of attachment of (**a**) BeWo (*n* = 5) or (**b**) JEG-3 trophoblast spheroids (*n* = 8) to a monolayer of RL-95 endometrial epithelial cells. Control: cells without treatment; msiRNA: cells transfected with 0.2 nM Silencer^®^ Select Negative Control siRNA; HO-1 siRNA: cells transfected with 0.2 nM Silencer^®^ Select HO-1 siRNA. Statistical analysis was performed using one-way ANOVA with the Bonferroni correction for multiple testing. *** *p* < 0.001.

**Figure 2 cells-13-00376-f002:**
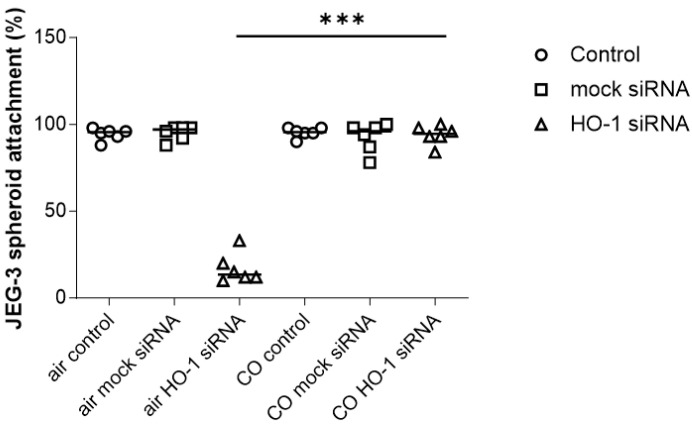
Percentage (%) of attachment of JEG-3 spheroids (*n* = 5) to a monolayer of RL-95 endometrial epithelial cells. Cells were incubated in normal conditions (air) or in the presence of 500 ppm CO. Control: cells without treatment; msiRNA: cells transfected with 0.2 nM Silencer^®^ Select Negative Control siRNA; HO-1 siRNA: cells transfected with 0.2 nM Silencer^®^ Select HO-1 siRNA. Statistical analysis was performed using one-way ANOVA with the Bonferroni correction for multiple testing. *** *p* < 0.001.

**Figure 3 cells-13-00376-f003:**
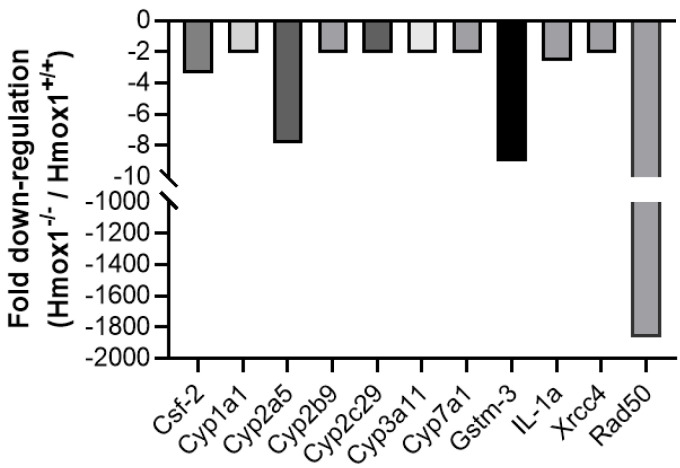
Fold downregulation of stress genes in a pool of *Hmox1^−/−^* uteruses (*n* = 7) when compared to a pool of *Hmox1^+/+^* uteruses (*n* = 8) using a PCR Array.

**Figure 4 cells-13-00376-f004:**
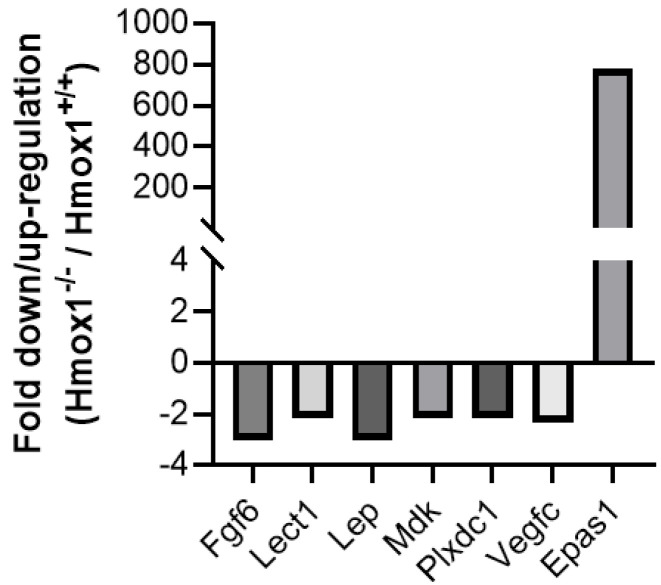
Fold regulation of angiogenesis genes in a pool of *Hmox1^−/−^* uteruses (*n* = 7) when compared to a pool of *Hmox1^+/+^* uteruses (*n* = 8) using a PCR Array.

**Figure 5 cells-13-00376-f005:**
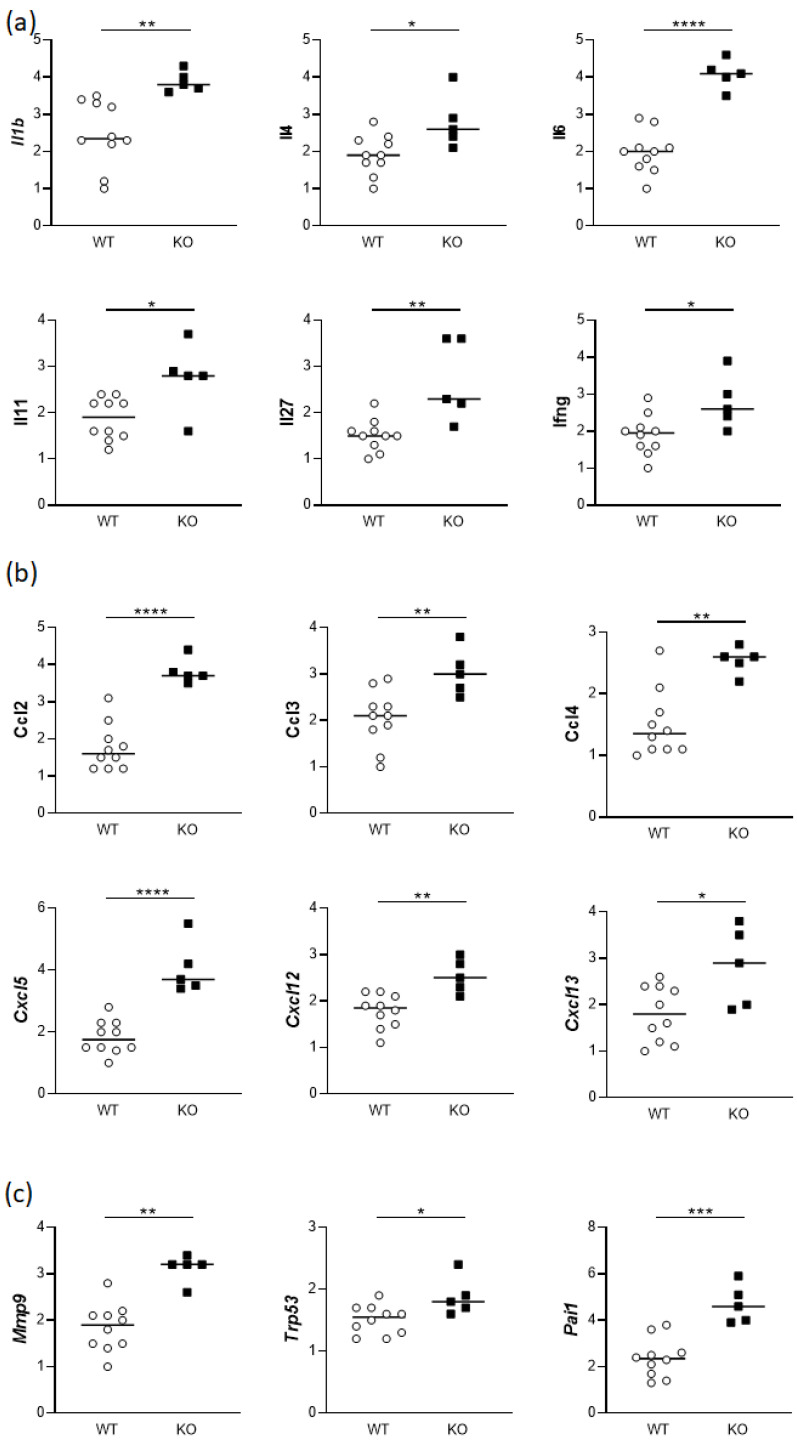
Gene expression of (**a**) cytokines, (**b**) chemokines, and (**c**) *Mmp9*, *Trp53*, and *Serpine1* in the uteruses of gd14 females. WT: *Hmox1^+/+^* mice (*n* = 10); KO: *Hmox1^−/−^* mice (*n* = 6). Gene expression was normalized to reference genes (*Actb*, *Gapdh*, *Rplp0*, *and Ubc*). Statistical differences were analyzed by unpaired t-test. * *p* <0.05; ** *p* <0.01; *** *p* < 0.001; **** *p* < 0.0001.

## Data Availability

The datasets generated and/or analyzed during the current study are available from the corresponding author on reasonable request.

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
