# Peer review of "Absence of Heme Oxygenase-1 Affects Trophoblastic Spheroid Implantation and Provokes Dysregulation of Stress and Angiogenesis Gene Expression in the Uterus"

_cells, 2024, doi:10.3390/cells13050376_

Round 1

Reviewer 1 Report

Comments and Suggestions for Authors

The authors conducted a beautiful study that observed that the reduction of HO-1 provokes a dysregulation in the expression of angiogenesis and stress-related genes in the uterus, leading to the development of a uterine inflammatory profile, which could decrease the rate of conception. The manuscript was very well written and conducted, so there are only a few minor points left to be improved.

- Reference 21 is a thesis, I suggest changing it to the published article originating from the thesis.

- Why was the reversal of HO-1 deficiency using CO2 only performed in JEG-3 spheroids? Did the authors not do it on another cell line or did they not get the expected result?

Reviewer 2 Report

Comments and Suggestions for Authors

In this article Zenclussen et al. studied the involvement of heme oxygenase 1 (HO-1) during implantation using two different experimental approaches. First, the authors evaluated the effect of CO treatment on the attachment of trophoblasts spheroids generated from placental cells where the expression of HO-1 was silenced by siRNA transfection. Additionally, the authors characterized the expression profile of different genes involved in angiogenesis and stress on Hmox1 wt and ko mice. The topic is interesting however, as it is presented now, the work has two unconnected parts. The in vitro results are not complemented by the in vivo results. The experiments performed with uterus samples from Hmox1+/+ and Hmox1-/- mice do not help to answer any question related to the in vitro findings. For this reviewer authors must perform in vivo experiments to study the effect of CO in uterus cells. On the other hand, it would be interesting to evaluate the expression of angiogenesis and stress-related genes in RL-95 cells subjected to CO treatment. Additionally, I do not agree with the selected cell lines for the generation of spheroids as JEG3 and BeWo cells derivate from tumors. Could the authors employ a cell line from first trimester? It is more appropriate to study implantation. Finally, there is no justification about the selected CO dose.   

Other concerns:

- The abstract must be improved. Specifically, the aim of the in vivo experiments is not clear.

- The bibliography is not up dated.

- The justification of the use of 500 ppm of CO must be included in the manuscript.

- Use “blastocyst-like spheroids” instead of “blastocyst” when referring to 3D cultures. I suggest to include this modification also in the title of the article.

- Why did the authors decide to continue the analysis using only JEG-3 cells? The justification must be included in the manuscript.

- Taking into account the pleiotropic effects of HO-1, it is surprising that CO treatment fully reversed the effect of HO-1 siRNA (Fig 2). Please, discuss this outcome.

- The discussion includes a lot of specific results (for example, lines 412-421) and in the last paragraph nothing is mentioned about the relevance of in vitro experiments.

Overall, I believe that the authors should improve some aspects of the presentation of their results to make this work suitable for publication in Cells.

Reviewer 3 Report

Comments and Suggestions for Authors

Comments for the authors: In this paper, the authors investigated the absence of heme oxygenase-1 affects blastocyst implantation and provokes a dysregulation of stress and angiogenesis gene expression in the uterus.
This study is well-designed, enjoyable, and contains novelties.

Minor Remarks: I recommend supplementing the discussion with the following: Please elaborate on how your results could be used in therapy or diagnostics?

Reviewer 4 Report

Comments and Suggestions for Authors

Absence of heme oxygenase-1 affects blastocyst implantation and provokes a dysregulation of stress and angiogenesis gene expression in the uterus

The authors presented a study supporting the hypothesis that HO-1 is essential for successful implantation of blastocysts. Knocking down HO-1 in trophoblastic spheroids impaired their ability to implant in endometrial epithelial cells. However, when spheroids were cultured in the presence of carbon monoxide (CO), one of the byproducts of the HO-1 reaction, the implantation capacity was reestablished. This suggested that CO may mediate the implantation process. The study also found that animals deficient in HO-1 have multiple disturbances, including reproductive problems, and that HO-1 is essential for successful implantation. The lack of HO-1 dysregulates other molecules in the uterine environment, contributing to implantation failure. The study also identified alterations in stress, angiogenesis, cytokine, and chemokine markers in the uterus of animals deficient in HO-1, further supporting the importance of HO-1 in implantation. Overall, the findings suggest a crucial role for HO-1 in early pregnancy stages and emphasize its significance in both placental and uterine functions.

Comments to the authors:

1. Introduction:

The authors should consider breaking down some lengthy sentences for improved clarity. Also, while the study's objectives are implied, explicitly stating the research objectives at the end of the introduction would provide a clear roadmap for the reader.

2. Materials and Methods

The passage number of tested cell lines and type of lysis buffer is not specified.

Provide details about how Hmox1+/+ and Hmox1-/- mice were genotyped to confirm their status.

Provide details about the internal controls and normalization strategies used in the assays.

Clarify whether the expression levels were normalized to the mean of the reference genes individually for each sample or for the pooled samples.

3. Results and discussion

Overall, while the content is insightful, enhancing the presentation of results will significantly improve the clarity of findings.

Suppl. Fig. 1: The separate cutting of WB bands, without providing the complete membranes raises many concerns. The results lack clarity due to the presentation of only one replicate. Including additional biological replicates data, complete (whole) WB membranes with molecular weight markers is crucial for both, HO-1 and beta-actin.

While the results mention that for example, the attachment was significantly impaired, it would be beneficial to provide quantitative data for results descriptions.

The section mentioning the reversal of the HO-1 knockdown effect by CO is very brief. Expanding on the experimental details would enhance the interpretation of these results.

Number of samples or mice used in the experiments is indicated. However, the authors do not provide the actual sample sizes for each experiment, which must be indicated and statistical analysis from at least 3 replicates has to be provided.

The discussion passage goes into detail about each gene's downregulation or upregulation. While this level of detail is essential for scientific rigor, ensure that the main findings and implications are better summarized.

With these improvements, the manuscript could be reconsidered for publication in Cells.

Comments on the Quality of English Language

The overall quality of English in the manuscript appears to be good, with clear and coherent sentences. However, there are some areas where sentence structure and clarity could be improved for better readability. Additionally, there are instances where more concise and direct language could be used.

Reviewer 5 Report

Comments and Suggestions for Authors

In this work entitled “Absence of heme oxygenase-1 affects blastocyst implantation and provokes a dysregulation of stress and angiogenesis genes expression in the uterus” authors study the implications of HO-1 and its byproduct carbon monoxide (CO) in blastocyst implantation. It examines the effects of HO-1 knockdown in cell cultures and the resulting changes in gene expression related to stress and angiogenesis in the uterus. The study  demonstrates that HO-1 deficiency impairs blastocyst attachment, which can be partially compensated by CO treatment. This highlights HO-1's crucial role in implantation and the possible contributions of its dysregulation to implantation failure.

However, there are several major and minor concerns that must be addressed:   

-How did authors choose to use 500 ppm of CO? Does it reflect what it is observed in vivo? At what extent does it compensate HO-1 loss? Is this concentration greater than the one that could be observed endogenously by HO1 activity?

- Improve Figure 1 resolution.

-In the trophoblastic spheroid generation and adhesion assay methodology should be clarified since it is not clear.  

-Only one cell line is assessed in Figure 2. The use of one cell line raises questions about the generalizability of the findings. Including additional cell lines or explain why the use of only one cell line.

-The siRNA knockdown approach's efficiency needs to be quantitatively presented, possibly including control treatments to validate the specificity of the siRNA for HO-1.

- paragraph including Line 313, is not complete and thus difficult to follow. What are authors comparing? Additionally circulating cytokines should not be measured by RNA expression.

-Unify style for gene names in Figure 4.

-Number of mice is not specified either in material and methods or the legend to figures 5, 6 and 7.

- Figures 5 6 and 7 should be merged.

- Theres is a lack of in vivo results reflecting the role of CO as described in the in vitro experiments. The study suggests carbon monoxide (CO) as a compensatory mechanism for the lack of HO-1. However, this is based on in vitro observations, and the actual compensatory pathways in vivo might be more complex and varied. Please address

-The manuscript claims a strong role of HO-1 in implantation based on in vitro models. More evidence or a discussion of how these findings translate to in vivo conditions would enhance the credibility of the conclusions.

- The manuscript should inlcude a more detailed statistical analysis, including data variability and replicates, to strengthen the validity of the conclusions drawn.

 -The study uses in vitro 3D culture models to mimic the implantation process. While these models offer valuable insights, they cannot completely replicate the in vivo environment of the uterus during implantation, which includes a complex interplay of cellular, molecular, and systemic factors, thus the manuscript should address the model limitations.

Comments on the Quality of English Language

-

Round 2

Reviewer 4 Report

Comments and Suggestions for Authors

The improvements in the introduction, materials, methods, and results have enhanced the clarity of the authors' findings. I appreciate the effort the authors have put into addressing concerns.

With these final adjustments, I believe the manuscript will be ready for publication in Cells.

Reviewer 5 Report

Comments and Suggestions for Authors

The authors have successfully addressed all the concerns raised.